# Modulation of Host Antiviral Innate Immunity by African Swine Fever Virus: A Review

**DOI:** 10.3390/ani12212935

**Published:** 2022-10-26

**Authors:** Wen-Rui He, Jin Yuan, Yu-He Ma, Cheng-Yan Zhao, Zhong-Yuan Yang, Yuhang Zhang, Shichong Han, Bo Wan, Gai-Ping Zhang

**Affiliations:** International Joint Research Center of National Animal Immunology, College of Veterinary Medicine, Henan Agricultural University, Zhengzhou 450046, China

**Keywords:** African swine fever virus, innate immune response, cGAS-STING pathway, JAK-STAT pathway, inflammatory response

## Abstract

**Simple Summary:**

Immune evasion is a prerequisite for successful viral infection, and viral proteins involved in regulating host antiviral innate immunity have always been the molecular targets for the development of novel African swine fever virus (ASFV) vaccines. To monitor potential virulent and immunoregulatory factors, we summarized the research progress on ASFV regulating host inflammatory and innate immune responses, which will be helpful for the research of vaccines and antiviral drugs against African swine fever (ASF).

**Abstract:**

African swine fever (ASF), caused by African swine fever virus (ASFV), is a highly contagious and fatal disease found in swine. However, the viral proteins and mechanisms responsible for immune evasion are poorly understood, which has severely hindered the development of vaccines. This review mainly focuses on studies involving the innate antiviral immune response of the host and summarizes the latest studies on ASFV genes involved in interferon (IFN) signaling and inflammatory responses. We analyzed the effects of candidate viral proteins on ASFV infection, replication and pathogenicity and identified potential molecular targets for novel ASFV vaccines. These efforts will contribute to the construction of novel vaccines and wonder therapeutics for ASF.

## 1. Introduction

African swine fever (ASF) is a highly contagious swine disease caused by African swine fever virus (ASFV), and it is characterized by a high fever, hemorrhaging, ataxia and depression [1,2]. After its first description in Kenya in 1921, ASF has broken out through Africa, Europe, South America, Latin America and Asia. ASF outbreaks have caused great economic losses and severely threatened the development of the pig industry worldwide. However, effective vaccines for ASF are not currently available. The prevention and control of ASF represents a major strategic need that must be addressed [3]. The pathogenesis of ASFV and the balance between the virus and its host directly determine the available strategies for ASF vaccine development. However, large gaps in the knowledge on viral proteins and the mechanisms responsible for regulating the host innate immune system have severely hindered further insights [4].

Nonetheless, a series of studies have been carried out on genetically engineered vaccines for ASF, including recombinant live attenuated vaccines (LAVs) and single-cycle vaccines [5]. BA71ΔCD2v, derived from the BA71 strain, lacks the CD2-like gene (CD2v), and it can protect the pigs from death after lethal challenge with its parental strain and heterologous strain [6]. HLJ/18-7GD, a seven-gene-deleted LAV, is fully attenuated in pigs and prevents the pigs from dying after lethal ASFV challenge [7]. However, reports on “field-isolated mutant virus” infections in pig farms in China have been increasing since early 2020. Due to the inconspicuous symptoms and relatively low viral titers of such infections, these attenuated ASFV strains with gene deletion or mutation are becoming a new source of pollution and infection, which greatly increases the complexity of ASF prevention and control in China [8]. Viral infection represents a battle and balance between the virus and its host, and the innate antiviral immunity is the first line of host defense against viral infection [9]. Thus, successful immune evasion is the first step. Interferon (IFN) and inflammatory responses, the major components of the antiviral innate immunity, play significant roles in ASFV infection and pathogenesis [10,11]. Consequently, viral proteins involved in immunoregulation are the main targets for antiviral agents and novel vaccines development; therefore, the mechanisms of ASFV immune evasion must be deciphered for the rapid and efficient development of ASF vaccines.

## 2. Overview of African Swine Fever Virus

Up to now, ASFV is the only reported member of the genus *Asfivirus* in the family *Asfarviridae* [12]. Similar to other nucleocytoplasmic large DNA viruses (NCLDVs), the genomes of this “normal” double-stranded DNA (dsDNA) virus vary in length from 170 to 190 kilobase pairs. Cytoplasm-replicated ASFV encodes more than 150 open reading frames (ORFs), which enable genome replication, transcription, translation, assembly, and immune escape [13,14]. Meanwhile, this “unique” large DNA virus ASFV possesses a five-layered structure and icosahedral morphology from other NCLDVs [1,15]. The core of viral particles is a nucleoid formed by the viral genome and proteins responsible for genome replication, transcription and repair. The core shell externally, which is formed from orderly arranged proteins, is encapsulated by the inner envelope and icosahedral capsid. The external envelope outmost forms from the plasma membrane during its budding [16].

Thus far, the available data have shown that ASFV mainly infects and replicates in porcine monocytes, macrophages, endothelial cells and renal tubular epithelial cells [17]. After invading cells via macropinocytosis or clathrin-mediated endocytosis [18,19], ASFV uncoating occurs, and it escapes from endosomes when its inner membrane fuses in the endosomal membrane. The viral core then migrates to the perinuclear viral factory through the microtubule system and initiates viral replication [20,21,22].

Studies confirmed that p72, p54, p30 and CD2v are major immunogens of ASFV. The inoculation of CD2v protein expressed in vitro may provide partial protection against infection by homologous virulent strains [23]. Certain proteins can induce neutralizing antibodies, including p30, p54, and p72, although they fail to protect pigs from virulent ASFV challenge. The deletion of DP71L, DP96R, B119L or DP148R significantly attenuated viral virulence and showed protective effects against infection by virulent ASFV strains [24,25]. Therefore, the biological functions of multiple ASFV-encoded proteins must be clarified to decipher the mechanism of proteins responsible for immune escape. Such findings will lay a solid theoretical foundation and provide insights for the development of novel ASF vaccines.

## 3. ASFV Regulation of Innate Immune Response

Antiviral innate immune and inflammatory responses are the first line of host defense against viral infection. These responses are initiated once the conserved viral structural components called pathogen-associated molecular patterns (PAMPs) are sensed by the pattern recognition receptors (PRRs) of host cells [26]. The sensing of viral PAMPs by PRRs activates signaling cascade, which is followed by the transcription and synthesis of antiviral effectors, including IFNs and proinflammatory cytokines [27,28]. These effector proteins play crucial roles in both innate and adaptive immune responses. For example, the maturation and activation of antigen-presenting cells are triggered by innate immunity and can further promote the adaptive immune system to effectively clear the invading viruses [29].

Nucleic acids are the most important viral PAMPs, and can be sensed by several PRRs, including members of the Toll-like receptor (TLR) family, RIG-I-like receptors (RLRs), and cyclic GMP-AMP synthase (cGAS). PRRs differ in their amino acid sequences, conformational structures and subcellular localizations, and these characteristics ensure that invading viruses can be sensed in a timely and effective manner [9,30]. Upon the binding of their distinct ligands, PRRs initiate downstream signaling events, including the recruitment of adaptor proteins, activation of kinases and transcription factors, which finally leads to the synthesis of antiviral effectors [31].

However, despite this extremely sophisticated antiviral immune system, ASFV has ravaged the pig industry worldwide for many years, which indicates that ASFV has created a relatively complete system of immune evasion that is on balance with its hosts. Multiple proteins encoded by the large genome of ASFV lay the foundation for effective infection, replication and complex immune escape. In recent years, a small number of ASFV proteins participated in the subversion of the host antiviral innate immunity have been identified. In this review, we mainly focus on the molecular mechanisms of the ASFV-regulated cGAS stimulator of IFN genes (STING), Janus kinase signal transducers and activators of transcription (JAK-STAT) signaling pathways, as well as inflammatory responses. We aim to summarize the research status of ASFV immune evasion and provide novel targets for the development of protective vaccines against ASF.

### 3.1. Regulation of ASFV on cGAS-STING Pathway

After internalization, the dsDNA of ASFV is mainly recognized by the cytoplasmic DNA sensor cGAS [16,32]. In uninfected cells, cGAS exists in the cytoplasm as monomers without DNA-binding or enzymatic activity [33]. After recognizing and binding DNA, cGAS catalyzes the synthesis of cyclic GMP-AMP (cGAMP) from ATP and GTP. As a second messenger, cGAMP in turn associates with and activates the adaptor STING, which further translocates to the ER-Golgi intermediate compartment (ERGIC) and recruits TBK1 and IRF3. Next STING and IRF3 are phosphorylated by TBK1, followed by the nuclear transport of dimerized IRF3 and the transcription of type I IFNs [34,35].

Studies have demonstrated that ASFV triggers the production of IFN-β at 4 dpi in pigs, and IFN treatment can significantly inhibit ASFV replication in vitro and in vivo [36,37], which not only provides a new strategy for the prevention of ASF, but also indicates the critical role of IFN in the pathogenesis of ASFV. The ASFV OUR T88/3 strain, which lacks partial genes from MGF360 or MGF505, induces higher levels of type I IFN than the parent strain [38,39], thus indicating that MGF360 and MGF505 members are involved in regulating IFN expression. Consistently, the IFN-β levels in ASFV Armenia/07 virulent strain infection cells are lower than those of uninfected cells, while the ASFV NH/P68-attenuated strain activates the cGAS-STING axis early after viral infection, followed by a high level of IFN-β expression [32].

Apart from differences in the regulation of IFN expression between different strains, members of the ASFV MGF family also participate in directly regulating the production of IFN. A276R, a member of MGF360, impairs IFN-β induction in an IRF3-dependent manner. However, the detailed process underlying this inhibition remains to be further explored [40]. MGF360-12L can significantly inhibit the poly (I:C)-triggered activation of the host immune response. Mechanistically, MGF360-12L impairs the interaction between importin α and NF-κB and inhibits the nuclear translocation of p65, leading to the reduced activation of host antiviral response [41].

Early-expressed MGF505-7R is an important inhibitor of the cGAS-STING pathway, and MGF505-7R-deficient ASFV induces a higher type I IFN production and displays relatively weaker virulence in piglets than wild-type ASFV. Mechanistically, MGF505-7R interacts with STING and impairs the expression of STING by upregulating the autophagy-related protein ULK1. In addition, MGF505-7R can interact with IRF3 and IKKα to block the nuclear translocation of IRF3 and NF-κB, which is followed by reduced IFN-β and proinflammatory cytokine production [42,43]. Moreover, MGF505-7R also functions by interacting with and degrading IRF7 and TBK1 via the autophagy, cysteine and proteasome pathways [44]. 

The ectopic expression of MGF505-11R disturbs cGAS-STING-mediated immune responses. MGF505-11R specifically binds to STING and degrades STING via the lysosomal, ubiquitin–proteasome and autophagy pathways [45]. Additionally, MGF360-11L inhibits IFN-I production by associating with TBK1 and IRF7, which is followed by the degradation of TBK1 and IRF7 via the cysteine, ubiquitin–proteasome and autophagy pathways [46]. MGF360-14L binds to E3 ubiquitin ligase TRIM21, and facilitates the K63-linked ubiquitination of IRF3, thereby destabilizing IRF3 and inhibiting the type I IFN production [47].

Along with MGF members, recent studies indicated that other ASFV proteins, especially structural proteins, also participate in regulating the cGAS-STING pathway. Overexpressed CD2v can significantly induce the NF-κB-dependent transcription of IFNs and downstream antiviral genes in PK15 cells or porcine peripheral blood mononuclear cells and macrophages, which can be counteracted by treatment with CD2v antibody or NF-κB inhibitor. Thus, ASFV may activate the NF-κB and IFN pathways via CD2v in porcine lymphocytes and macrophages [48].

The capsid proteins of ASFV, E120R and M1249L are involved in regulating ASFV immune evasion. E120R inhibits cGAS-STING-mediated immune response by targeting IRF3. The phosphorylation of IRF3 is impaired due to the binding of E120R to its C-terminal domain, which disturbs the recruitment of IRF3 to TBK1, thus leading to the reduced production of IFNs [49]. M1249L is also associated with IRF3, which significantly inhibits the cGAS-STING pathway by inducing IRF3 degradation via the lysosomal pathway [50]. S273R, an ASFV core shell protein, is also involved in regulating the expression of IFN and ISGs. S273R interferes with the association between IKKε and STING, and it functions in an enzyme-catalytic-activity-dependent manner [51]. In addition, S273R also works by inhibiting the expression of the antiviral effector FoxJ1, which mainly functions by targeting ASFV MGF505-2R and E165R and activates cGAS-STING-initiated immune responses [52]. ASFV inner envelope protein E248R inhibits cGAS-STING-mediated innate immunity by inhibiting STING expression, while the detailed mechanism remains unclear [53].

Interestingly, EP364R and C129R, two proteins encoded by ASFV, are homologous with nuclease. They can interact with and cleave cGAMP in a phosphodiesterase activity-dependent manner, thus subverting cGAMP-mediated IFN signaling. Moreover, EP364R competitively combines with cGAMP, leading to the blockage of the interaction between cGAMP and STING, further enhancing its ability to antagonize the host antiviral innate immune response [54]. I226R can impair antiviral responses through the suppression of NF-κB and IRF3 activation. An ASFV mutant with the deleted I226R gene is dramatically attenuated and able to protect the pigs from death due to challenge with a homologous virus [55,56].

Studies have found that the attenuated A137R-deficient ASFV can provide protection against its parental strain ASFV Georgia 2010 challenge [57]. Compared with Georgia 2010, the A137R-deficient ASFV induces robust expression of type I IFN in primary porcine alveolar macrophages (PAMs). In-depth research shows that A137R interacts with TBK1 and promotes the autophagy-mediated lysosomal degradation of TBK1, thus leads to the inhibition of cGAS-STING-mediated immune response [58]. Current research also demonstrates that the ASFV early-transcription gene DP96R and late-transcription gene E2 ubiquitin-binding enzyme I215L can inhibit the production of type I IFN by targeting TBK1 [59,60]. Early-expressed E301R inhibits the activation of cGAS-STING pathway by interacting with IRF3 and impairing the nuclear translocation of IRF3 [61], while the roles of DP96R, I215L and E301R in ASFV-infected pigs need to be further studied.

Taken together, in addition to MGF family, multiple proteins of ASFV are involved in determining the virulence of ASFV (Figure 1). Meanwhile, it is can be easily observed that more than half of the studies are performed at the protein level, the mechanisms of which remain unclear. Vaccines lacking one or several genes may be attenuated to some extent and provide partial protection. In other words, vaccines that simply delete MGF family members or candidates could not provide pigs with a virulent viral challenge. Therefore, other ASFV-encoded proteins playing vital roles in the immune escape of ASFV need to be urgently identified, as this will be of great significance for the construction of new vaccines against ASF.

### 3.2. Regulation of ASFV on JAK-STAT Pathway

The first line of defense against viral infections is mediated by IFNs, which can be grouped into three types: type I IFN (mainly IFN-α/β), type II IFN (IFN-γ) and type III IFN (IFN-λ) [62,63,64]. The canonical JAK-STAT pathway is initiated by the binding of IFNs to their cognate cell surface receptors, which are then brought into close spatial proximity with JAKs (JAK1, JAK2, JAK3, and TYK2), phosphorylate IFN receptors, and STATs (STAT1 and STAT2). Next, phosphorylated STAT1 and STAT2 interact with IRF9 to form the heterotrimer IFN-stimulated gene factor (ISGF) 3. Subsequently, ISGF3 enters the nucleus to initiate ISG transcription [10,65]. Many ISGs restrict the spread of viruses by directly targeting virus infection and replication, such as transcription and translation, to effectively eliminate invading viruses [66,67].

It has been well-studied that replication of ASFV is notably impaired by treatment with IFNs. In turn, infection with both virulent and attenuated strains interferes the expression of ISGs induced by IFN though targeting the JAK/STAT pathway. Viral infection impairs the nuclear translocation of the ISGF3 complex, and causes the degradation of STAT2 and cleavage of STAT1, further suggesting the importance of inhibiting this pathway for successful viral replication. Thus, insights into the IFN antagonistic properties of ASFV may be helpful for exploring new strategies to eliminate this plaguing disease [68,69].

As a multifunctional protein, MGF505-7R also antagonizes the activation of IFN downstream signaling. MGF505-7R can promote the degradation of JAK1 and JAK2 by upregulating the expression of E3 ubiquitin ligase RNF125 and inhibiting the expression of Hes5, respectively, thus leading to the inhibition of the IFN-induced JAK-STAT signaling pathway activation. Compared with wild-type ASFV, MGF505-7R-deficient ASFV induces higher levels of IRF1 in both PAMs and swine. The replication ability, virulence and pathogenicity of mutant viruses are significantly reduced [70]. MGF360-9L interacts with STAT1 and STAT2 and inhibits the activation of IFN signaling pathway by inducing the degradation of STATs via the ubiquitin–proteasome pathways. The virulence of 360-9L-deficient ASFV is partially weakened, which indicates that MGF360-9L is one key virulence gene of ASFV [71].

Moreover, I215L is also involved in subverting JAK-STAT signaling pathway. Mechanistically, I215L interacts with IRF9 for autophagic degradation in a ubiquitin-conjugating activity-independent manner [72]. Altogether, the antagonistic ability of ASFV against IFN is closely related to its pathogenicity, and further exploration of ASFV immune evasion strategies in the cGAS-STING and JAK-STAT pathway will facilitate the cognization on ASFV pathogenesis, and provide insights into the construction of novel vaccines against ASF. For example, MGF505-7R may be a promising novel target for the development of ASFV-attenuated vaccines based on its central role in immune evasion and pathogenesis of ASFV.

### 3.3. Regulation of ASFV on Inflammatory Response

In addition to IFNs and ISGs, ASFV infection can also induce the secretion of the proinflammatory cytokines, the tumor necrosis factor (TNF)-α, IL-1β, IL-6, and IL-8 [73]. Previous studies demonstrated that elevated levels of TNF-α play important roles in the pathogenesis of ASF owing to its proinflammatory, proapoptotic and procoagulant profile [14,74]. As major components of host antiviral innate immunity, TNF-α and IL-1β play central roles in the control of the replication of many viruses, such as the classical swine fever virus [75] and Japanese encephalitis virus [76]. NF-κB activation is a hallmark following TNFα and IL-1β stimulation. Once the plasma membrane-anchored receptors (TNF receptor 1 (TNFR1) or IL-1 receptor (IL-1R)) bind to their respective ligands, they recruit and activate adaptors and kinases, including TNFR-associated death domain protein (TRADD), myeloid differentiation primary response 88 (MyD88), TGF-β-activated kinase 1 (TAK1), TAK1-binding protein (TAB) 2 and TAB3, and IκB kinase (IKKβ) [77,78]. Following the phosphorylation and degradation of IκBα, the transcription factor NF-κB is released and translocated to the nucleus, thus leading to the production of various cytokines and subsequent inflammatory responses [79,80].

Recently, several ASFV proteins that regulate NF-κB activation have been identified (Figure 2). The early expressed protein L83L specifically associates with IL-1β; however, the deletion of the L83L gene only slightly affects the virulence of ASFV [81]. Moreover, the early-expressed A238L is a homolog of NF-κB inhibitor IκB, and it can inhibit NF-κB activation and regulate the expression of proinflammatory cytokines in cells. However, in vivo and in vitro studies have shown that A238L-deficient viruses exhibit inapparent differences in the growth characteristics, virulence, and pathogenicity to its parental strains [82,83,84,85].

I215L is also involved in modulating inflammatory responses. Ectopic expression of I215L inhibits NF-κB activation by targeting IKKβ or its upstream responses [86]. Moreover, F317L is also associated with IKKβ and suppresses its activation, which subsequently leads to the enhancement of IκBα stabilization and blocks the activation of NF-κB. This results in the decreased expression of various proinflammatory cytokines and increased viral replication efficiency [87]. The infectious progeny viral titers of H240R-deficient ASFV are significantly reduced. Importantly, H240R affects ASFV assembly and recombinant ASFV infection induces high levels of expression of inflammatory cytokines in PAMs. This suggests that H240R may be involved in the antagonism of the host inflammatory responses [88].

In summary, the deletion of a single ASFV gene involved in the inflammatory response does not appear to affect viral virulence and pathogenicity, indicating that multiple proteins encoded by ASFV are related to the viral-infection-induced inflammatory response, and they function in a coordinated manner. Further in-depth studies on the mechanisms underlying the regulation of inflammatory responses by ASFV are urgently needed and may be helpful for understanding this devastating disease.

## 4. Conclusions

Vaccines are the primary tools for the prevention and control of viral diseases in animals. Pigs that recover after infection with ASFV may be resistant to this disease, which provides the possibility for ASF vaccines. However, clear breakthroughs have not been made in this area despite several attempts, including inactivated vaccines, recombinant subunit vaccines, DNA vaccines, and live attenuated vaccines. To date, no commercial ASF vaccines have been developed. The major difficulty of ASF vaccine development is the limited understanding of the molecular mechanisms of complex ASFV proteins and host antiviral responses. 

As a first-line of host defense against viral infection, the innate immune response is characterized by rapid, non-specific and universal responses. IFNs plays crucial roles in preventing the rapid propagation and spread of ASFV in the early stages of infection. Proinflammatory cytokines, including TNF-α and IL-1β, can be robustly induced after viral infection. They can be sensed by their respective receptors and induce the production of downstream inflammatory genes, thus leading to the activation of inflammatory responses. However, studies show that ASFV infection significantly impairs the activation of host antiviral responses. 

ASFV encodes more than 150 ORFs, and previous studies have shown that multiple proteins, including A238L, DP96R, MGF360, MGF505-7R and A276R, are involved in promoting viral replication by regulating the innate immune and inflammatory responses during ASFV infection. Nevertheless, studies have proved that a single-candidate gene deletion hardly affects the virulence and pathogenicity of ASFV. Undoubtedly, characterizing the immune evasion and pathogenesis of ASFV is only the first step. Comprehensively elucidating the delicate balance between the regulation of the innate immune response by ASFV and the pathogenic process of viral infection represents an important theoretical basis for the development of ASF vaccines. The deletion of several or a small number of ASFV key regulatory genes may be an effective strategy for novel ASF vaccine construction.

## Figures and Tables

**Figure 1 animals-12-02935-f001:**
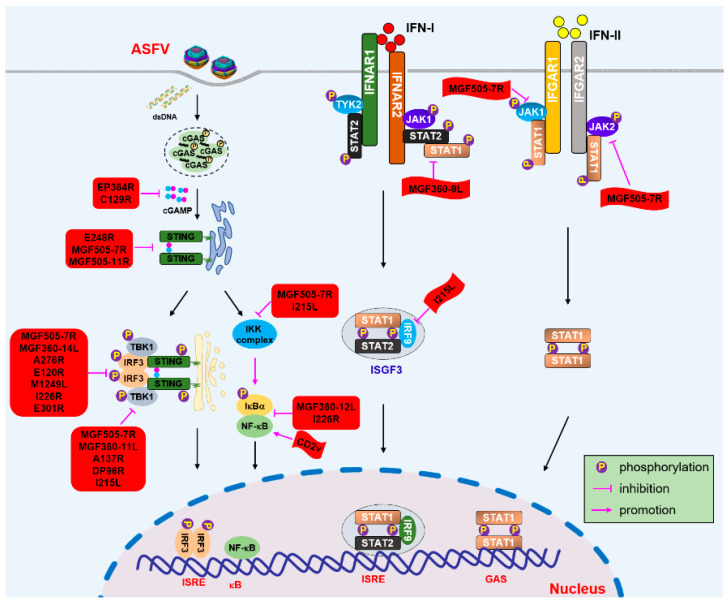
Molecular mechanisms and regulation of cGAS-STING and IFN signaling.

**Figure 2 animals-12-02935-f002:**
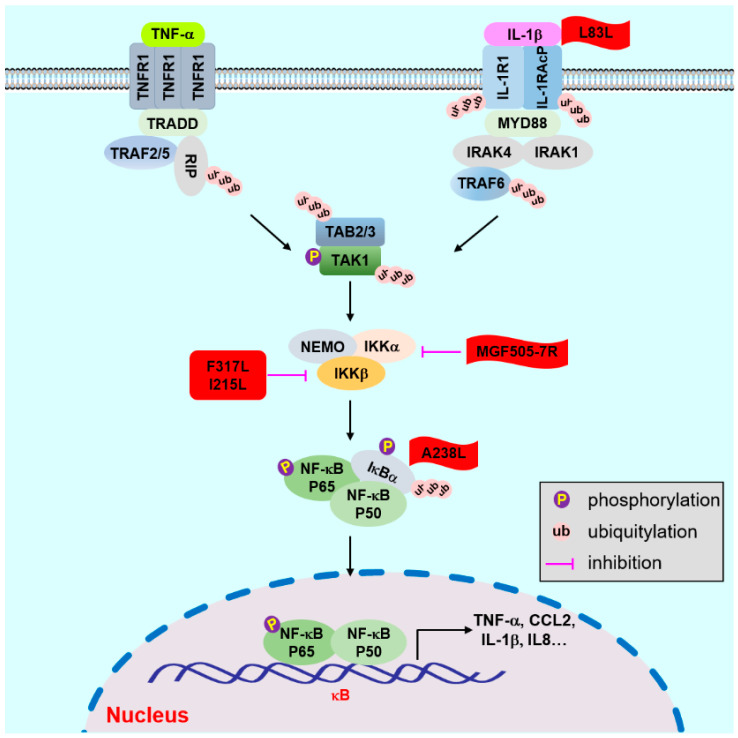
Molecular mechanisms and regulation of TNF-α/IL-1β-triggered signaling.

## Data Availability

This article is a literature review; therefore, all data described here were obtained from the research cited in the references.

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
