# Peer review of "Modulation of Host Antiviral Innate Immunity by African Swine Fever Virus: A Review"

_animals, 2022, doi:10.3390/ani12212935_

Round 1

Reviewer 1 Report

Immune evasion is a prerequisite for successful viral infection, and viral proteins involved in regulating host antiviral innate immunity have always been the molecular targets for the development of novel ASFV vaccines. To monitor potential virulent and immunoregulatory factors, this review summarized research progress on ASFV regulating host inflammatory and innate immune responses, which will be helpful in the development of novel vaccines against ASF.

1,  The figures should be improved in presentation. All figure legends should be supplemented in detail, for example, which icon represents phosphorylation or ubiquitination? What do the different arrows mean?

2,  Recently, several other ASFV proteins regulating host innate immunity were reported, please update the latest information in this review, such as E301R (10.1016/j.virusres.2022.198931).

3,  The references need to be checked, and the format should be consistent, for example, references 22, 31, 37, Journal of Virology should be corrected to J Virol.

Author Response

Response to Reviewer 1 Comments

Point 1: The figures should be improved in presentation. All figure legends should be supplemented in detail, for example, which icon represents phosphorylation or ubiquitination? What do the different arrows mean?

Response 1: We thank the reviewer for these suggestions. Detailed information for the figures has been added in this version of the manuscript.

Point 2: Recently, several other ASFV proteins regulating host innate immunity were reported, please update the latest information in this review, such as E301R (10.1016/j.virusres.2022.198931).

Response 2: We thank the reviewer for this suggestion. Actually, several other proteins involved in regulating host innate immunity are not included in the previous versions of the manuscript. We have now updated our data to Oct. 18, for example, ASFV E301R (page 5, line 200-202) and the new mechanism of MGF505-7R (page 4, line 153-155) have been added in this version

Point 3: The references need to be checked, and the format should be consistent, for example, references 22, 31, 37, Journal of Virology should be corrected to J Virol.

Response 3: We thank the reviewer for this suggestion. We have now checked that all references are relevant to the contents of the manuscript and corrected formats of several references (references 6, 14, 18, 23, 32, 38, 49, 54, 56, 57, 62, 79, 84, 87).

Reviewer 2 Report

Well written review paper, but outside my area of expertise so I am not able to judge the adequacy of the review of the current literature.  I am intrigued by mention of the use of unnatural gene-deleted "wild vaccine viruses" being used on pig farms in China, which represents a development of considerable concern.

Author Response

Response to Reviewer 2 Comments

Point 1: Well written review paper, but outside my area of expertise. so I am not able to judge the adequacy of the review of the current literature. I am intrigued by mention of the use of unnatural gene-deleted "wild vaccine viruses" being used on pig farms in China, which represents a development of considerable concern.

Response 1: We thank the reviewer for this kindly evaluation to our manuscript. We have now corrected it to “field-isolated mutant virus” (page 2, line 47), which will be more precise than the statement “wild vaccine viruses” mentioned in the previous versions. Besides derived from the use of vaccines, these gene-deleted or mutant viruses may also be the result of the continuous adaptation and evolution of ASFV. So, continuous surveillance is required and benefited for the better understanding of viral evolution and further improved the implementation of prevention and control measures. On the other hand, no matter what are the origins of these mutant viruses, it is urgent to development more safe and effective vaccines for the pig industry all around the world, especially China.

Reviewer 3 Report

The review "Research progress on African swine fever virus regulating host antiviral innate immunity" provides a comprehensive review of recent research on the mechanisms utilised by AFSV to subvert the interferon system in pigs. The English is generally excellent although phrases such "Thus successful immune evasion is the first shot in this game." (pg 2) are probably unnecessary. There is very little in the way of critical assessment of the research that underpins the mechanisms described. Are they all confirmed or is there disagreement on them or whether they occur within an infected pig?

Two minor points are the need for explanation of the nomenclature of ASVF protein coding orfs, can the authors provide a table or equivalent listing the named proteins and explaining their function or predicted function. Secondly, when describing a mechanism, consider a little more detail. For example "MGF505-7R can promote the degradation of JAK1 and JAK2 by upregulating the expression of E3 ubiquitin ligase RNF125...". How exactly does this protein do this? Is it a transcription factor and when is it expressed?

Finally, the Prospect section could enlarge on how this knowledge will enhance vaccine design. Are the authors suggesting whole-sale deletions of the ASFV genome or is a more nuanced approach possible through removal of a small number for key regulatory genes?  

Author Response

Response to Reviewer 3 Comments

Point 1: The review "Research progress on African swine fever virus regulating host antiviral innate immunity" provides a comprehensive review of recent research on the mechanisms utilised by AFSV to subvert the interferon system in pigs. The English is generally excellent although phrases such "Thus successful immune evasion is the first shot in this game." (pg 2) are probably unnecessary. There is very little in the way of critical assessment of the research that underpins the mechanisms described. Are they all confirmed or is there disagreement on them or whether they occur within an infected pig?

Response 1: We thank the reviewer for these suggestions. We have corrected "Thus successful immune evasion is the first shot in this game" to "Thus, successful immune evasion is the first step" (page 2, line 53). Actually, little critical assessment was made in the previous versions, we have tried to evaluate these candidates objectively basing on the previous studies to further address the reviewer’s concerns (page 4, line 143, 183; page 5, line 204-211; page 6, line 229, 239-241). However, as described at page 5, line 204-211, more than half of the studies are performed at the protein level, and mechanisms of which remains unclear or nebulous. Therefore, further in-depth studies are urgently needed, and other ASFV-encoded proteins involved in the immune evasion of ASFV, also need to be identified. These works will be of great significance for the construction of novel efficient vaccines against ASF.

Point 2: Two minor points are the need for explanation of the nomenclature of ASVF protein coding orfs, can the authors provide a table or equivalent listing the named proteins and explaining their function or predicted function. Secondly, when describing a mechanism, consider a little more detail. For example "MGF505-7R can promote the degradation of JAK1 and JAK2 by upregulating the expression of E3 ubiquitin ligase RNF125...". How exactly does this protein do this? Is it a transcription factor and when is it expressed?

Response 2: We thank the reviewer for these suggestions. Firstly, the table listing the ASFV proteins and their functions are available in an excellent study lately (reference 14), and we have now cited it in this manuscript (page 2, line 67) to facilitate the readers to get this information more easily. Secondly, according to the data at present, more detailed description of the mechanisms has been added in the revised manuscript. For example, as one of the most thoroughly studied ASFV proteins, the functions and mechanisms of MGF505-7R were introduced systematically (page 4, line 147-155; page 6, line 233-240). However, the results indicate that MGF505-7R interacts with RNF125 and Hes5, followed by upregulating RNF125 expression and downregulating Hes5 protein. The precise mechanisms of MGF505-7R involved in the transcription, translation or degradation of RNF125 and Hes5, remain unknown until now.

Point 3: Finally, the Prospect section could enlarge on how this knowledge will enhance vaccine design. Are the authors suggesting whole-sale deletions of the ASFV genome or is a more nuanced approach possible through removal of a small number for key regulatory genes?  

Response 3: We thank the reviewer for these suggestions. LAVs have been proved to provide complete protection against the homologous virus, which would be the next generation of ASF vaccines. While targeted genes for deletion or mutation remain to be elusive. As we know, this is due to the complex nature of the virus, especially the incomplete understanding of immune protection determinants of ASFV. On the other hand, previous studies have shown that that the single gene deletion in ASFV hardly affects the virulence and pathogenicity in pigs. Undoubtedly, characterizing the immune evasion and pathogenesis of ASFV is only the first step. Comprehensively elucidating the delicate balance between the regulation of the innate immune response by ASFV and the pathogenic process of viral infection will provide an important theoretical basis for the development of ASF vaccines. Deletion of several or a small number of ASFV key regulatory genes may be an effective strategy for novel ASF vaccine construction.
